# Childhood trauma, PTSD/CPTSD and chronic pain: A systematic review

**Maria Karimov-Zwienenberg**[1,2]*, **Wilfried Symphor**[2], **William Peraud**[2], **Greg Décamps**[2]

**1** Centre Hospitalier Agen-Nérac, Agen, France, **2** UR 4139 Laboratoire de Psychologie, Université de Bordeaux, Bordeaux, France

* zwienenbergm@ch-agen.nerac.fr

## Abstract

### Background

Despite the growing body of literature on posttraumatic stress disorder (PTSD) and chronic pain comorbidity, studies taking into account the role of childhood exposure to traumatic and adverse events remains minimal. Additionally, it has been well established that survivors of childhood trauma may develop more complex reactions that extend beyond those observed in PTSD, typically categorized as complex trauma or CPTSD. Given the recent introduction of CPTSD within diagnostic nomenclature, the aim of the present study is to describe associations between childhood trauma in relation to PTSD/CPTSD and pain outcomes in adults with chronic pain.

### Methods

Following PRSIMA guidelines, a systematic review was performed using the databases Pubmed, PsychInfo, Psychology and Behavioral Sciences Collection, and Web of Science. Articles in English or French that reported on childhood trauma, PTSD/CPTSD and pain outcomes in individuals with chronic pain were included. Titles and abstracts were screened by two authors independently and full texts were consequently evaluated and assessed on methodological quality using JBI checklist tools. Study design and sample characteristics, childhood trauma, PTSD/CPTSD, pain outcomes as well as author's recommendations for scientific research and clinical practice were extracted for analyses.

### Results

Of the initial 295 search records, 13 studies were included in this review. Only four studies explicitly assessed links between trauma factors and pain symptoms in individuals with chronic pain. Findings highlight the long-term and complex impact of cumulative childhood maltreatment (e.g., abuse and neglect) on both PTSD/CPTSD and chronic pain outcomes in adulthood.

### Conclusion

This review contributes to current conceptual models of PTSD and chronic pain comorbidity, while adding to the role of childhood trauma and CPTSD. The need for clinical and

**Data Availability Statement:** All relevant data are within the manuscript.

**Funding:** The author(s) received no specific funding for this work.

**Competing interests:** The authors have declared that no competing interests exist.

translational pain research is emphasized to further support specialized PTSD/CPTSD treatment as well as trauma-informed pain management in routine care.

## Introduction

Over the past two decades, the comorbidity between chronic pain (i.e., persistent pain >3 months) and post-traumatic stress disorder (PTSD) has been well established [1–3]. PTSD is a psychiatric diagnosis based on the presence of a set of specific symptoms (e.g., flashbacks, hypervigilance, avoidance) that might occur after experiencing or witnessing a life-threatening event such as a disaster or assault. A recent meta-analysis including 21 studies reported higher PTSD prevalence up to 57% in individuals with chronic pain compared to 2–9% in the general population [4]. In the context of pain management, this alarming comorbidity represents many challenges as it has been associated with higher levels of pain severity [5], pain disability [6], and opioid use [7]. Furthermore, individuals with chronic pain and comorbid PTSD typically report increased levels of PTSD severity, emotional distress and psychiatric comorbidity than controls [8–10].

Several conceptual frameworks have been proposed, such as shared vulnerability and mutual maintenance models suggesting the interplay of neurobiological, emotional and cognitive factors involved in comorbidity [2, 11, 12]. Despite different hypotheses of causality and interaction, the particular nature of the relationship between chronic pain and PTSD remains uncertain. Depending on the studied population or condition, pain could both contribute to and maintain PTSD. Similarly, PTSD has been considered an important risk factor in the development of chronic pain when compared to controls [13].

Studies agree however that a history of adverse childhood events may be associated with both PTSD and chronic pain in adulthood [14–16]. Childhood adversity typically includes experiences of abuse, neglect as well as exposure to household dysfunction, parental psychopathology and early parental loss [17]. There is cumulative systematic and meta-analytical evidence demonstrating increased risk of chronic pain and pain-related disability in individuals reporting single or cumulative exposure to adverse childhood events, in particular maltreatment (e.g., childhood abuse, neglect) [15, 18, 19]. Although psychological distress has been identified as a key aspect to this phenomenon, few studies examined the role of PTSD in this context, indicating a gap in clinical and translational pain research, particularly in regard to trauma-informed pain management [20] as well as psychological treatment for comorbid trauma and chronic pain [21].

Additionally, it has been well established that survivors of childhood adversity may develop more complex and multifaceted reactions that extend beyond those observed in PTSD. These reactions have been commonly categorized as complex trauma or complex PTSD (CPTSD) [22, 23]. CPTSD describes the widespread and long-lasting consequences following exposure to ongoing and often inescapable interpersonal traumatic stress that occurs within the context of a significant relationship (e.g., childhood abuse, intimate personal violence) [22, 24]. Disparate adaptations to interpersonal trauma were initially conceptualized as an associated feature of PTSD by Disorders of Extreme Stress Not Otherwise Specified (DESNOS) [25]. However, due to the lack of sufficient evidence to support its inclusion as a unique diagnostic entity, DESNOS was eventually dropped from the fifth version of the Diagnostic and Statistical Manual (DSM) [26]. More recently, the World Health Organization published the 11[th] version of the International Classification of Diseases (ICD-11) [27] introducing CPTSD for the first

time into diagnostic nomenclature. Alongside the crucial presence of PTSD symptoms, the current model shares many similarities with DESNOS, including affect dysregulation, negative self-concept and interpersonal difficulties which are typically referred to as disturbances in self-organization (i.e., DSO symptoms) [28, 29]. Additionally, consistent with recent data [30, 31] and earlier conceptual research [29, 32], current ICD-11 guidelines expanded trauma exposure definition for PTSD and CPTSD by taking into account different types of interpersonal trauma, including childhood neglect and emotional abuse, in addition to DSM criterion A events. In the context of chronic pain, there is some preliminary evidence suggesting worsened pain outcomes in survivors of childhood abuse with CPTSD as opposed to PTSD symptoms alone [33]. As PTSD and CPTSD are currently considered related disorders, it seems of timely interest to address how these relate to pain chronicity in order to promote effective treatment options and pain management for individuals with comorbid PTSD/CPTSD and chronic pain.

## Objectives

Despite the growing body of research on the trauma-chronic pain relationship, evidence in relation to PTSD/CPTSD following childhood exposure to traumatic or adverse events remains scarce. The aim of this study is to conduct a systematic review exploring existing data on the described links, while taking into account authors' recommendations for future research and clinical practice. For the purpose of this review, in line with previous conceptual research and current ICD-11 PTSD/CPTSD guidelines, the term childhood trauma is used to address the exposure of traumatic or adverse events before the age of 18 years.

Specifically, this review seeks to describe in individuals with chronic pain:

1. Prevalence and characteristics of trauma factors, including:

   a. Childhood trauma

   b. Posttraumatic stress symptomatology, including PTSD and CPTSD symptoms.

2. Characteristics of the relationship between childhood trauma, PTSD/CPTSD and chronic pain symptoms

   a. Relationship between childhood trauma and posttraumatic stress symptomatology, including PTSD and CPTSD.

   b. Relationship between trauma factors and chronic pain symptoms

3. Authors' recommendations regarding:

   a. Scientific research

   b. Clinical practice

## Methods

### Search strategy

Before conducting this systematic review, a search in the Prospero database showed that, to our knowledge, no literature review is currently in progress on this subject (https://www.crd.york.ac.uk/PROSPERO/ accessed on July 2023).

To conduct the present systematic review, we followed the guidelines described by the Preferred Reporting Items for Systematic Reviews and Meta-Analysis (PRISMA) [34]. A search

**Table 1. Search strategy terms.**

*Search terms for chronic pain*
("chronic pain" OR "persistent pain" OR "fibromyalgia" OR "chronic pelvic pain" OR "chronic low back pain" OR "chronic back pain" OR "chronic migraine" OR "chronic headache" "chronic widespread pain")
AND
*Search terms for PTSD/CPTSD*
("post-traumatic stress" OR "posttraumatic stress" OR "post traumatic stress" OR "PTSD" OR "complex PTSD" OR "dissociation" OR "stress disorder")
AND
*Search terms for childhood trauma*
("childhood trauma" OR "childhood abuse" OR "childhood maltreatment" OR "adverse childhood experiences" OR "childhood neglect" OR "interpersonal trauma" OR "childhood victimization" OR "early life trauma")

was performed from 1st of August 2023 using the following databases: Pubmed (Medline); PsychInfo (EBSCO*host*), Psychology & Behavioral Sciences Collection (EBSCO*host*), and Web of Science (Web of Knowledge). Search strategy terms are presented in Table 1.

## Inclusion and exclusion criteria

As per guidance, PICOTS framework [35] was used to structure the review process by defining selection criteria as follows: [1] Population, [2] Intervention, [3] Comparison, [4] Outcome, [5] Time and [6] Setting. Predefined inclusion and exclusion criteria are presented in Table 2.

## Study selection and data extraction

Studies were selected independently by two authors (MKZ and WS) by screening titles and abstracts in systematic review. The selected studies were then subject to full text screening by applying the selection criteria. Reasons were documented during the process. In case of disagreement, discrepancies were adjudicated by a third author (WP) until a consensus was reached among the three authors. Once study eligibility was confirmed, data was extracted between September 2023 and December 2023 by one author (MKZ) which was then verified

**Table 2. Study eligibility using PICOTS framework.**

| PICOTS Element | Inclusion criteria | Exclusion criteria |
|---|---|---|
| **Population (P)** | Adults (≥18yrs) suffering from chronic pain, including: • Population with clinical diagnostic established prior to study (e.g., clinical sample) • Population with self-reported chronic pain (presence of persistent pain > 3 months) (e.g., community sample) | 1. Chronic pain does not appear as a study sample characteristic or is not explicitly reported according to a clinical diagnosis or self-reported measure. (Reason 1) |
| **Intervention (I)** | Present study objectives do not include assessment of intervention efficacy. However, observational studies are included as well as interventional studies evaluating psychotherapy's efficacy | 2. Medical or pharmaceutical studies, including efficacy trials. (Reason 2) |
| **Comparator (C)** | With or without comparative sample. | |
| **Outcome (O)** | Studies reporting on all of the following outcomes: • Chronic pain symptomatology • Posttraumatic stress symptomatology, including PTSD and CPTSD • Childhood trauma (<18yrs) | 3. Chronic pain symptomatology does not appear in study results. (Reason 3) 4. PTSD/CPTSD symptoms do not appear in study results (Reason 4) 5. Childhood trauma does not appear in study results. (Reason 5) |
| **Timing (T)** | No specific timing | |
| **Setting (S)** | Empirical peer-reviewed articles published in English and/or in French. | 6. Theoretical articles and reviews. (Reason 6) 7. Study is published in another language than English or French. (Reason 7) 8. Non-peer reviewed articles or non-published studies. (Reason 8) |

by a second author (WS). The following items were identified for data collection: authors, year, country, study design, study sample, chronic pain condition, chronic pain symptomatology, childhood trauma exposure, PTSD/CPTSD, interaction data between trauma factors and chronic pain symptomatology, and finally, author's recommendations for scientific research and clinical practice.

### Critical appraisal of study quality

The methodological quality of each included study was independently assessed by two researchers (MKZ and WS) using the corresponding design-specific critical appraisal checklist tools provided by the Joanna Briggs Institute (JBI) [36]. The following JBI critical appraisal checklist tools were used for this review: case control studies, analytical cross-sectional studies, quasi-experimental, as well as cohort studies. Each component was rated as "Yes", "No", "Unclear, or "Not Applicable". If needed, discrepancies were discussed between reviewers or by consulting a third author (WP) until consensus was reached. Based on previous systematic reviews [37, 38], studies with a JBI score higher than 70% were considered as high quality, those with scores between 50% and 70% as moderate quality, and those with a score less than 50% as low quality.

## Results

### Study design and participants characteristics

The initial search returned 297 records, of which 36 were retained for full-text analysis. Finally, 13 articles [39–50] were included in this systematic review without disagreement (i.e., inter-judge agreement = 100%). Fig 1 presents a flow-diagram of the research article selection process.

The 13 studies included in this review were published between 2005 and 2023 and conducted in Europe (Italy, $n = 1$; Belgium, $n = 1$; Spain, $n = 2$; Germany, $n = 1$), Turkey $n = 1$; Israel ($n = 3$), and the US ($n = 4$). Four were case-control studies, 7 cross-sectional studies, 1 quasi-experimental study and 1 cohort study. There was some variety in sample sizes across the studies, ranging from 70 to 295 participants, recruited both from clinical ($n = 9$) and community settings ($n = 4$). All study populations compromised exclusively ($n = 7$) or predominantly female participants (>64%). Finally, Fibromyalgia (FM) was found to be the most studied pain condition ($n = 9$), followed by unspecified chronic pain ($n = 3$), and Interstitial cystitis/bladder pain syndrome (IC/BPS) ($n = 1$). In terms of missing data, it was found that the majority of the included studies did not address all outcomes of interest to this review. Unreported information on outcomes was identified as "Not Reported" (N/R). Study findings are listed in Tables 3 and 4.

### Quality of the included studies

The results of the quality assessment are summarized in Table 5. Quality appraisal using the JBI checklist tools indicated overall moderate to high quality studies. Nine studies scored above 70% [39, 40, 42–44, 46–49], three studies scored between 50% and 70% [41, 45, 51], and the remaining one study [50] scored 13%. The main limitations of the single low-quality study were lack of objective and valid methods of assessment regarding chronic pain outcomes.

### Study objective 1: Descriptive data of trauma factors in individuals with chronic pain

**a. Childhood trauma.** All but one study [47] included in this review reported childhood trauma in terms of maltreatment, demonstrating higher prevalence [39, 41, 42] and severity

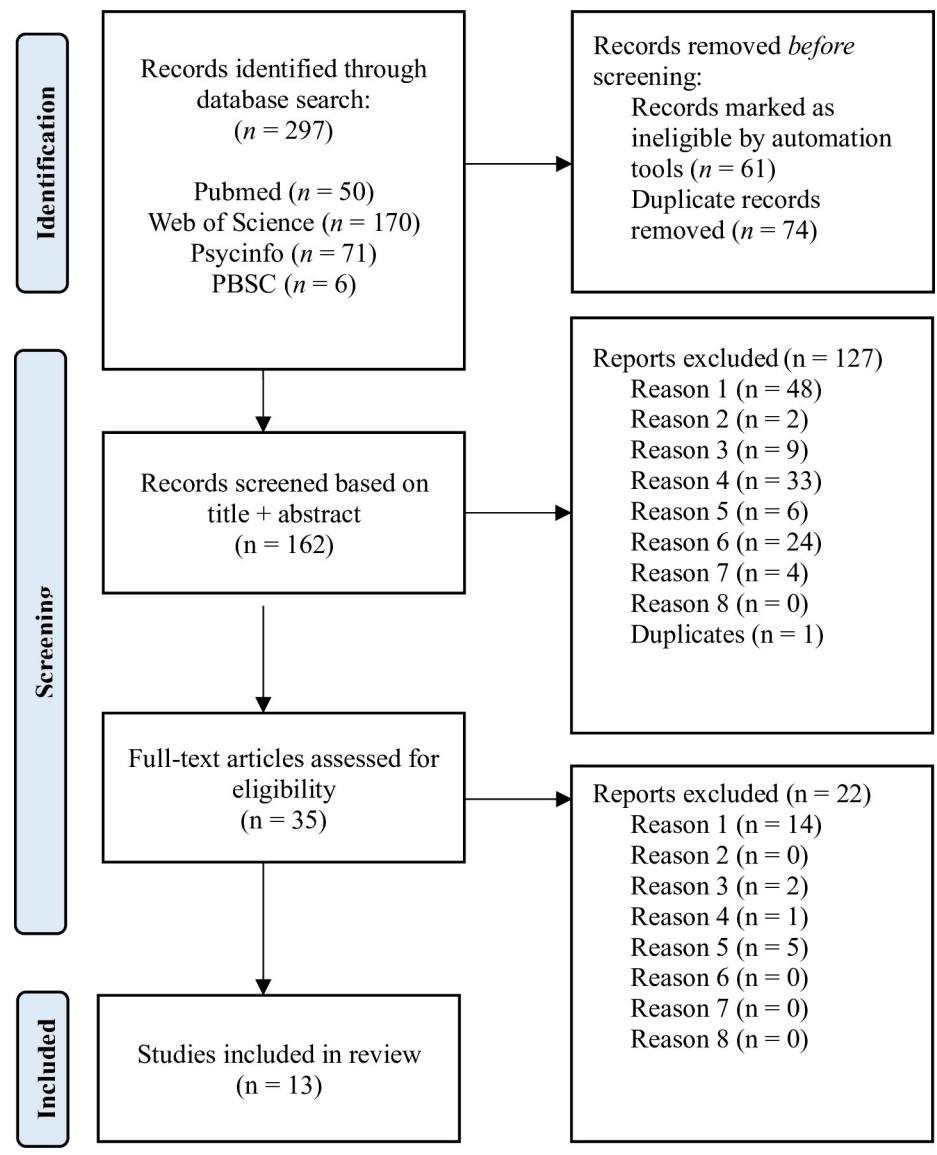

**Fig 1. PRISMA flow diagram of systematic search results.**

[45, 48] for emotional abuse and neglect compared to other forms of childhood maltreatment in individuals reporting chronic pain. In addition, a cohort study [48] demonstrated significative interrelations between all types of abuse and neglect, except for sexual abuse and neglect in a clinical sample of FM patients. Ciccone et al. [40] found no differences in childhood physical or sexual abuse between women reporting FM and healthy controls.

When compared with other medical conditions, studies found higher childhood maltreatment rates and severity in individuals with chronic pain, in particular with regards to neglect [39, 41, 45], sexual abuse [39, 41], and emotional abuse [41, 45].

Only two studies assessed childhood trauma exposure based on PTSD qualifying stressors following DSM criteria [42, 47]. For example, Gardoki-Souto et al. [42] found that most prevalent traumatic events were reported during childhood compared to adulthood. Physical, sexual, and emotional abuse were identified as most commonly reported traumatic events during

**Table 3. Study characteristics, descriptive and interaction data.**

| Study | Population (% female) | CP symptoms (measure) | Trauma factors Characteristics | | Trauma factors Descriptive data | | Trauma factors—CP Interaction data | |
|---|---|---|---|---|---|---|---|---|
| | | | CT (measure) | PTSD (measure) | CT | PTSD/CPTSD | CT - PTSD/CPTSD | b. Trauma factors - CP symptoms |
| **Clinical sample** | | | | | | | | |
| Alciati et al., 2020 Italy [39] | 30 PFM (96.7%) 40 SFM (92.5%) | Pain intensity (VAS) FM impact (FIQ) | Prevalence of abuse & neglect (CTQ) | PTSD prevalence (SCID-5) | Highest prevalence for emotional neglect both in PFM (93.3%) and SFM (90%). Higher prevalence in physical neglect (90% vs 65%)* and sexual abuse (20% vs 0%)** between PFM and SFM. | Prevalence between 20.8% (PFM) and 22.5% (SFM), no group differences. | Physical neglect is an independent risk factor for PMS. | N/R |
| Coppens et al., 2017 Belgium [41] | 159 FM/CWP 83 FD 53 Achalasia (100%) | Pain severity (MGPQ) | Prevalence, severity, presence and number of abuse and neglect (CTQ) | PTSD prevalence (PTSD-ZIL) | Highest prevalence for emotional neglect in FM/CWP, FD, and Achalasia (19.6% - 31.4%). Higher prevalence for CT overall (49.4%)***, including sexual abuse (21.4%)*, emotional abuse (24.7%)* and emotional neglect (31.8*) in FM/CWP vs Achalasia. | Higher PTSD prevalence in FM/CWP (26%) vs FD (4.9%)*** and Achalasia (12.2%)***. Higher PTSD severity in FM/CWP vs FD*** and Achalasia***. FM/CWP were about 5–7 times more likely to meet PTSD-criteria vs FD*** and achalasia***. | Higher rates of PTSD (34% vs 17%)* and PTSD severity*** in FM/CWP reporting CT compared to FM/CWP without CT. Dose response relationship between the number of CT (3+ vs none) and PTSD severity*. Dose response relationship between CT severity and PTSD severity**. | No relationship between CT severity and quantitative or qualitative pain severity in FM/CWP patients. Relationship between PTSD severity and both quantitative* and qualitative pain*** severity in FM/CWP patients (medium effect sizes). Indirect effect of CT severity on both qualitative and quantitative pain reports through PTSD severity. No moderation effect of PTSD on the relationship of CT and pain severity. |
| Gardoki-Souto et al., 2022 Spain [42] | 88 FM (100%) | Pain severity (VAS) Pain disability (PDI) FM impact (FIQ) | Prevalence and severity of abuse and neglect (CTQ) Self-reported traumatic or stressful experiences, including childhood (0-15y) (thematic analysis) | PTSD prevalence (EGEP-5) Level of current subjective perturbation in relation to traumatic event identified by EGEP-5 (SUD) | Highest prevalence for emotional abuse (63.6%) and neglect (62.5%). The most prevalent traumatic events to occur between 0 and 15 years were physical abuse (n = 39), sexual abuse (n = 32), and emotional abuse (n = 29). | PTSD prevalence 71.5%. 34 women selected a traumatic event that occurred in childhood The average subjective distress score in relation to traumatic event was 8.43 (from 0 to 10, sd = 0.26) | Most prevalent traumatic events categorized by age occurred during childhood and adolescence. Almost half of the sample (34/81) reported childhood trauma as most significant traumatic event. | Emotional abuse, emotional neglect and physical neglect predicted FM-related pain impact** and disability**. Higher scores of physical neglect predicted higher levels of pain intensity. |

(*Continued*)

**Table 3.** (Continued)

| Study | Population (% female) | CP symptoms (measure) | Trauma factors Characteristics | | Trauma factors Descriptive data | | Trauma factors—CP Interaction data | |
|---|---|---|---|---|---|---|---|---|
| | | | CT (measure) | PTSD (measure) | CT | PTSD/CPTSD | CT - PTSD/CPTSD | b. Trauma factors - CP symptoms |
| Häuser et al., 2015 US + Germany [44] | 141 FM with polysymptomatic distress (score >11). US sample: 71 German sample: 70 (95.8%) | FM severity (PSD) Years since onset of FM/ CWP diagnosis (medical questionnaire) Pain disability (PDI). | Severity and prevalence of "severe and very severe" abuse and neglect (CTQ) | PTSD prevalence (M-CIDI; PDS) | No differences in CT severity between FM patients from US vs Germany. | PTSD prevalence: 33.8%, no differences between group | FM patients with PTSD reported higher prevalence of "severe and very severe" CT vs FM patients without PTSD. PTSD was not correlated with CA severity only. | N/R |
| Hellou et al., 2017 Israel [45] | 75 FM (86.7%) 23 RA (87%) | FM severity (WIP; SSS) Pain disability (PDI) | Severity and prevalence of "severe and very severe" abuse and neglect (CTQ) | PTSD prevalence (PDS) | higher CT severity for emotional abuse**, emotional neglect** and physical neglect* in FM vs RA. Higher CT denial in RA vs FM*** | PTSD prevalence**: 37.3% in FM vs 8.7 in RA | N/R | N/R |
| López-López et al., 2021 Spain [46] | 36 FM FM+PTSD: 18 FM-PTSD: 18 38 HC (100%) | PPT PPTo (electric algometer) | Total severity score for abuse and neglect (CTQ) | PTSD diagnosis (established prior to study) | N/R | N/R | Higher CT severity* in FM+ PTSD patients compared to controls (FM-PTSD and HC) | FM patients showed lower basal PPT*** and PPTo*** vs HC. FM patients showed hypo reactivity under stress conditions, reflected by the lack (FM+PTSD) or delay (FM-PTSD) of a hyperalgesic response. Similar pain intensity and chronicity levels in FM+PTSD vs FM-PTSD. |
| Nicolson et al., 2010 US [48] | 70 FM/OA FM sample: 35 OA sample: 35 (100%) | Pain intensity (0–100 scale) | Prevalence above clinical cut-off scores, and severity of abuse and neglect (CTQ) | Lifetime history and recent PTSD (SSAGA-II) | Highest prevalence above clinical cut-off for emotional abuse (44%), followed by physical abuse and neglect (33%), sexual abuse (25%) and emotional neglect (20%). Subscale scores of abuse and neglect were intercorrelated***, except for sexual abuse and physical neglect. No differences between CT scores in FM vs OA. | Total sample: Lifetime history PTSD: 19% Recent PTSD: 11.4% | More severe childhood trauma was associated with a lifetime diagnosis of PTSD**. | Non significant trends for correlations between CT and daily pain intensity (.07) |

*(Continued)*

**Table 3.** (Continued)

| Study | Population (% female) | CP symptoms (measure) | Trauma factors Characteristics | | Trauma factors Descriptive data | | Trauma factors—CP Interaction data | |
| | | | CT (measure) | PTSD (measure) | CT | PTSD/CPTSD | CT - PTSD/CPTSD | b. Trauma factors - CP symptoms |
|---|---|---|---|---|---|---|---|---|
| Semiz et al., 2014 Turkey [51] | 56 FM (92.9%) 46 RA (87%) | FM impact (FIQ) | Severity of abuse and neglect (CTQ) | PTSD prevalence (PDS) | Higher CT severity in FM vs RA*. | Higher PTSD rates in FM (10.7%) vs RA (0%). | N/R | Positive correlation between FM impact and CT**. FM impact was higher in FM patients with PTSD vs without PTSD* |
| Peles et al., 2016 Israel [49] | 76 MMT MMT/SA: 68 MMT/SA + CP: 41 25 SATC SATC+CP: 20 (100%) | Self-reported chronic pain (> past 6 months) Time of onset, duration of pain, and severity (CPQ) | Age of onset sexual abuse (population characteristic) | CPTSD prevalence (SIDES-NOS-SR) Sexual-abuse related PTSD prevalence (CAPS) | Differences*** in age of onset for sexual abuse between SATC (mean age: 9; ± 6.3) vs MMT/SA (mean age 12.9; ± 4.5) | MMT/SA sample: 32.5% sexual-related PTSD, 19.1% CPTSD SATC sample: 84% sexual-related PTSD, 60% CPTSD | N/R | Both CP subgroups combined: Positive correlation* between pain severity and number of painful body regions. Positive correlation between pain severity, sexual-abuse related PTSD*** and CPTSD* severity. Positive correlation between number of painful body regions, sexual-abuse related PTSD** and CPTSD*** severity***. Negative correlation between age of onset of sexual abuse and pain duration*. No associations were found for PTSD, CPTSD, and age of onset of sexual abuse. |
| **Community sample** | | | | | | | | |
| Ciccone et al., 2005 US [40] | 50 FM/MDD 53 HC/MDD (100%) | Tender point count severity (standardized examination) Pain severity of present, usual, and highest pain, physical disability, number of pain locations, pain duration (GCPS) | Prevalence of childhood physical and sexual abuse (structured interview following Leserman's protocol) | PTSD cluster and diagnostic prevalence (PTSD checklist) | No group differences in sexual and/or physical abuse in childhood in FM/MDD (43.3%) vs HC/MDD (37.7%) | Higher PTSD prevalence in FM/MDD (26.9%) compared to HC/MDD (3.8%) ** after controlling for depression. Higher PTSD cluster rates for Intrusion (30.8 vs 9.4%) ** and Arousal (53.9% vs 22.6%)**, but not Avoidance in FM/MDD compared with HC/MDD. | N/R | Considering lifetime physical and sexual abuse (including childhood), only rape during adulthood was associated with increased risk of FM/MDD, mediated by PTSD**. |

(*Continued*)

**Table 3.** (Continued)

| Study | Population (% female) | CP symptoms (measure) | Trauma factors Characteristics | | Trauma factors Descriptive data | | Trauma factors—CP Interaction data | |
|---|---|---|---|---|---|---|---|---|
| | | | CT (measure) | PTSD (measure) | CT | PTSD/CPTSD | CT - PTSD/CPTSD | b. Trauma factors - CP symptoms |
| Hart-Johnson & Green, 2012 US [43] | 183 CP (64%) | Pain perception (MGPQ) Pain disability (PDI) | Prevalence of molestation, penetration and physical abuse (DAQ) | Pain-related PTSD prevalence (PCPT) | Higher physical childhood abuse in men vs women**, and in black CP compared with white CP* Highest rates for childhood penetration in black female CP compared with white female CP. Scores for men did not differ from either set of women's scores. White female CP reported lowest scores of physical abuse compared vs black men*, black women*, white men**. | N/R | Childhood molestation predicted pain-related PTSD only in male participants with CP*; controlling for race, sex and education. Male survivors of childhood molestation reported highest pain-related PTSD scores, while men without abuse history reported lowest pain-related PTSD scores. Female survivors of childhood molestation were equally likely to have pain-related PTSD as those without abuse history. | Childhood physical abuse** and penetration** predicted affective pain. Molestation in childhood was associated with higher affective pain* only in male CP. Sexual penetration in childhood predicted higher pain-related disability**. Abuse was not associated with sensory pain, and miscellaneous pain was related only to abuse in adulthood**. |
| McKernan et al., 2019 US [47] | 137 CP (75.9%) 64 IC/BPS (87.5%) | Pain intensity (0–10 scale) Widespread pain (Michigan Body Map) Central Sensitization (CSI) | Total CT score (CATS) | PTSD severity (PCL-5) | No differences in total CT rates in CPP vs IC/IBS, including sexual abuse. Women reported higher rates of childhood neglect*, while men reported higher rates of more general disaster/traumas* | PTSD prevalence in IC/BPS: 42% vs non IC/BPS CP: 23% Higher PTSD severity in IC/BPS vs CPP**. | Higher rates of childhood trauma exposure*** (large effect), including sexual abuse** (medium effect) in IC/BPS + PTSD vs IC/BPS -PTSD. No differences in adult trauma exposure, including physical/sexual trauma in IC/BPS + PTSD vs IC/BPS -PTSD. | Higher rates of current pain** (medium effect) and CS*** (large effect) in IC/BPS + PTSD vs IC/BPS only. Patients with co-occurring IC/BPS and CS reported higher levels of current and widespread pain***, PTSD symptoms*** vs IC/BPS without CS (large effect), as well as higher rates of childhood trauma* and lifetime sexual and physical abuse* (medium effect) |

*(Continued)*

**Table 3.** (Continued)

| Study | Population (% female) | CP symptoms (measure) | Trauma factors Characteristics | | Trauma factors Descriptive data | | Trauma factors—CP Interaction data | |
|---|---|---|---|---|---|---|---|---|
| | | | CT (measure) | PTSD (measure) | CT | PTSD/CPTSD | CT - PTSD/CPTSD | b. Trauma factors - CP symptoms |
| Tsur, 2023 Israel [50] | 160 survivors of childhood sexual abuse with CP: Pain-flashback sample: 37 Non-pain flashback sample: 58 No flashback sample: 65 (100%) | CP areas (heat map) CP intensity (MGPQ) | Child physical, sexual and emotional abuse severity (CTQ) | PTSD/CPTSD symptoms (ITQ) | A part from sexual abuse, the majority of participants also reported other types of CT exposure, including physical abuse (43.75%) and emotional abuse (25.63%). | Higher levels of PTSD in pain flashback vs non-pain flashback**; in non-pain flashbacks vs no-flashbacks***. Higher levels of DSO in pain flashback vs non-pain flashback**, and between pain flashback and no flashback***. | Higher physical abuse severity in pain flashback vs no flashback*. Higher sexual abuse severity in pain flashback vs non-pain flashback*, and in pain-flashback vs no flashback***. Higher emotional abuse severity in pain flashback vs no flashback***, and between non-pain flashback and no flashback***. | Highest chronic pain prevalence in pain flashback (51.4%) vs non-pain flashback (39.7%), and no flashback (18.8%)***. Higher chronic pain intensity in pain flashbacks vs non-pain flashbacks and no flashback***. Higher pain prevalence during sexual abuse in pain flashback (81.1%) vs non-pain flashback (39.7%), and no flashback (23.4)***. Higher pain intensity during traumatic event in pain flashback vs non-pain flashback***, and no-flashback***. Pain during sexual abuse was associated with higher risk of chronic pain prevalence in adulthood**, but not with chronic pain intensity. Chronic pain locations included pain-flashbacks body locations and wide-spread areas. |

*p < .05; ** p < .01; *** p < .001; CT = Childhood Trauma; CP = Chronic Pain; FM = Fibromyalgia; PFM = Primary FM; SFM = Secondary FM; CWP = Chronic Widespread Pain; FD = Functional Disorder; RA = Rheumatoid Arthritis; CPP = Chronic Pain Patients; IC/BPS = Interstitial Cystitis/Bladder Pain Syndrome; OA = Osteoarthritis; MMT = Methadone Maintenance Treatment; SA = Sexual Abuse; SATC = Sexual Abuse Treatment Center; MDD = Major Depressive Disorder; HC = Healthy Controls; VAS = Visual Analogue Scale; FIQ = Fibromyalgia Impact Questionnaire; MGPQ = McGill Pain Questionnaire; PDI = Pain Disability Index; PSD = Polysymptomatic Distress; WIP = Widespread Pain Index; SSS = Symptom Severity Scale; PPT = Pressure Pain Thresholds; PPTo = Pain Pressure Tolerance; CSI = Central Sensitisation Inventory; CPQ = Chronic Pain Questionnaire; GCPS = Graded Chronic Pain Scale; CTQ = Childhood Trauma Questionnaire; SCID-5 = Structured Clinical Interview for DSM-5; PTSD-ZIL = PTSD Zelf Inventarisatie Lijst; EGEP-5 = Evaluación Global de Estrés Postraumático– 5; PDS = Posttraumatic Diagnostic Scale; SIDES-NOS-SR = Structured Interview for Disorders of Extreme Stress-Non Other Specified-Self-Reported; CAPS = Clinician Administered PTSD Scale; PCL-5 = PTSD Checklist for DSM-5; DAQ = Drossman Abuse Questionnaire; SSAGA-II = Semi-Structured Assessment for the Genetics of Alcoholism; PCPT = Posttraumatic Chronic Pain Test; ITQ = International Trauma Questionnaire.

childhood. McKernan et al. [47] demonstrated differences in gender, with higher rates of childhood neglect observed in women, while men seemed to report more general disaster/trauma [47].

Finally, Hart-Johnson & Green [43] identified confounding effects of race and sex showing higher physical abuse under the age of 14 in male participants with chronic pain as opposed to women reporting chronic pain, with highest rates of abuse reported in black male participants

**Table 4. Author's recommendations for future research and clinical practice.**

| Study | Study objective 3: Author's recommendations | |
|---|---|---|
| | **Scientific research** | **Clinical practice** |
| Alciati et al., [39] | N/R | N/R |
| Ciccone et al., [40] | Future research should conduct in particular prospective studies to assess how rape and posttraumatic stress are implicated in the etiology of FM. | N/R |
| Coppens et al., [41] | N/R | Systematic screening for PTSD symptoms in patients with FM/CWP, by using semi-structured interviews or well-validated self-report questionnaires.<br>Both prevention and intervention or psychotherapeutic strategies should target PTSD symptoms and their impact on pain severity and general functioning. |
| Gardoki-Souto et al., [42] | Future research should assess trauma focused-interventions in FM and clarify the trauma-based etiology of FM in comparison to other functional somatic syndromes, medically unexplained symptoms, somatic symptoms, and related disorders following the DSM-V. | Systematic screening for psychological trauma and provide trauma-focused therapies within established multidisciplinary health care professionals, following existing FM guidelines. |
| Hart-Johnson & Green, [43] | More research is needed including men when addressing abuse in the study of chronic pain.<br>As present findings reveal a positive relationship between education and abuse history, future studies should address this issue in order to better understand educational differences CPPs reporting experiences of abuse.<br>Future studies should further investigate the role of other psychiatric diagnoses in order to add meaning to present findings. | Findings emphasize the need to screen all chronic pain patients for abuse regardless of race, age, or gender while pursuing effective treatment strategies to improve overall health and well-being.<br>Specific training for clinicians is recommended as barriers (e.g., lack of time, discomfort with subject, or lack of familiarity with the role of abuse) may hinder appropriate and effective screening methods. |
| Häuser et al., [44] | Future research on the etiology and pathophysiology of FM should consider concomitant mental disorders and psychological distress. | Considering the high prevalence of potential mental disorders in relation to abuse and trauma and their negative impact on FM outcome, appropriate screening for mental disorders is needed. |
| Hellou et al., [45] | Future research should look into the role of childhood adversity and denial in FM patients, including less violent aspects such as neglect in the etiology of chronic pain. | N/R |
| López-López et al., [46] | More research is needed to replicate present findings and test whether some psychological states of detachment (e.g., dissociation) might explain absence of pain response as a coping strategy in FM patients, based on Eccleston's model of a tripartite system of threat protection. The differences in FM patients based on the presence of PTSD should be considered in research studies following a differential profile approach. | If these results are replicated, implications for treatment might include intervention techniques based on Eccleston's model of a tripartite system of threat protection, helping FM patients with and without PTSD to engage in a more adaptive stress response. |
| McKernan et al., [47] | Replication and extension of current findings, particularly in terms of PTSD prevalence, is needed using larger clinical samples. Future investigations may explore the role of PTSD from childhood trauma in IC/BPS symptom development and maintenance, using structured diagnostic tools. Clinical trials may further investigate the effect of PTSD symptom reduction on pain management in similar chronic pain conditions. | Implementing established principles of trauma-informed care in the treatment, emphasizing patient-provider trust and rapport, reducing anxiety, and increasing perceived control during appointments:- initially assessing PTSD through monitoring specific symptoms over direct questioning (e.g., hyper-arousal, nightmares).<br>• encouraging collaboration and predictability during appointments in trauma survivors and enhancing safety as applied to patient examination procedures.<br>As PTSD appears to be associated with the "widespread" pain phenotype, multimodal treatment should be considered in these patients. |
| Nicolson et al., [48] | Future research should further evaluate childhood maltreatment in FM and other chronic pain conditions in order to clarify the origin and etiological significance of HPA axis dysregulation, and therefore better inform the development of interventions to reduce the deleterious effects of childhood maltreatment.<br>Additional research is needed to identify mediating or moderating factors on the relations childhood maltreatement and HPA axis function, namely comorbid syndromes or specific characteristics of CT (e.g., developmental timing, type), as well as subsequent experiences of violence or abuse, and ongoing interpersonal relations. | N/R |

*(Continued)*

**Table 4.** (Continued)

| Study | Study objective 3: Author's recommendations | |
|---|---|---|
| | **Scientific research** | **Clinical practice** |
| Peles et al., [49] | Future studies evaluating chronic pain and the issue of hyperalgesia among patients receiving treatment for opiate addiction should address sexual abuse history specifically. | Clinicians treating survivors of sexual abuse should ask specifically about, and pay attention to, complaints of chronic pain, in order to foster tailored adequate approaches in comprehensive treatment. |
| Semiz et al., [51] | Considering the impact of early trauma on HPA axis dysregulation, the relationship of trauma and FM should be further investigated using hormones and biomarkers. | Identification and treatment for PTSD, alexithymia and somatoform dissociative symptoms in FM patients are important, using trauma-focused approaches such as Eye Movement Desensitization and Reprocessing (EMDR). |
| Tsur [50] | More research, using longitudinal design, should clarify whether pain flashbacks play a role in the link between trauma and later chronic pain. Future studies should compare the role of pain flashbacks to other pain-related posttraumatic stress symptoms (e.g., avoidance of trauma-related pain sensations, pain-related hypervigilance). Various methodologies and contextual variation (including men, other age groups, and cultures) should be used while investigating pain flashbacks. The role of peritraumatic pain for later chronic pain warrants further investigation, exploring the link between physical injury during trauma and higher pain risk later in life. In the study of childhood abuse, CPTSD and pain outcomes, more attention should be addressed to somatic manifestations of CPTSD, in general, and pai n flashbacks, in particular. | N/R |

and lowest in white female participants. Sexual penetration during childhood was found to be most prevalent among black female participants when compared with male or white female participants.

**b. Posttraumatic stress symptomatology: PTSD/CPTSD.** In this review, the majority of the included studies described PTSD prevalence exclusively for predominantly female FM study samples with rates ranging from 10.7% to 37% [39–41, 44, 45, 48, 51]. One study [42]

**Table 5. Methodological quality of included studies.**

| Cohort study | Q1 | Q2 | Q3 | Q4 | Q5 | Q6 | Q7 | Q8 | Q9 | Q10 | Q11 | Quality score/11 |
|---|---|---|---|---|---|---|---|---|---|---|---|---|
| Nicolson et al., [48] | Y | Y | U | Y | Y | NA | Y | Y | Y | Y | Y | 82% |
| **Case control** | **Q1** | **Q2** | **Q3** | **Q4** | **Q5** | **Q6** | **Q7** | **Q8** | **Q9** | **Q10** | | **Quality score/10** |
| Ciccone et al., [40] | Y | Y | Y | Y | Y | Y | Y | U | Y | Y | | 90% |
| Coppens et al., [41] | N | N | N | Y | Y | Y | Y | Y | Y | Y | | 70% |
| Hellou et al., [45] | N | Y | Y | Y | Y | Y | N | Y | Y | N | | 70% |
| Semiz et al., [51] | Y | N | Y | Y | Y | Y | N | Y | Y | N | | 60% |
| **Quasi-experimental study** | **Q1** | **Q2** | **Q3** | **Q4** | **Q5** | **Q6** | **Q7** | **Q8** | **Q9** | | | **Quality score/9** |
| Lopez-Lopez et al., [46] | Y | N | Y | Y | Y | Y | Y | Y | Y | | | 89% |
| **Cross sectional** | **Q1** | **Q2** | **Q3** | **Q4** | **Q5** | **Q6** | **Q7** | **Q8** | | | | **Quality score/8** |
| Alciati et al., [39] | Y | Y | Y | Y | Y | Y | Y | Y | | | | 100% |
| Gardoki-Souto et al., [42] | Y | Y | Y | Y | Y | Y | Y | Y | | | | 100% |
| Hart-Johnson & Green, [43] | N | Y | Y | U | Y | Y | Y | Y | | | | 75% |
| McKernan et al., [47] | Y | Y | Y | Y | Y | N | Y | Y | | | | 88% |
| Tsur [50] | N | N | Y | N | N | N | U | N | | | | 13% |
| Peles et al., [49] | Y | Y | Y | Y | Y | Y | Y | Y | | | | 100% |
| Häuser et al., [44] | Y | Y | Y | Y | Y | N | Y | Y | | | | 88% |

Y = Yes; N = No; U = Unclear; NA = Not Applicable.

reported PTSD prevalence up to 71% following exposure to cumulative trauma as categorized by age. Results showed that most prevalent traumatic events occurred during childhood but continued into adulthood in the form of both different and recurrent types of events favoring a process of continuous re-traumatization. The lifelong impact of childhood trauma was further emphasized by high levels of current perceived distress in relation to past experiences of early life adversity.

When compared to controls, multiple studies showed higher PTSD prevalence and severity in individuals with chronic pain versus other medical conditions, including rheumatoid arthritis (RA) [45], functional disorders and achalasia [41], as well as healthy individuals [40]. Only one study [40] investigated PTSD symptom clusters, and found significantly higher rates for Intrusion and Arousal clusters, but not Avoidance when comparing a community sample of women with FM to healthy controls. Groups did not differ in childhood exposure to physical and/or sexual abuse.

Two studies included CPTSD measures in addition to PTSD [49, 50] providing evidence for CPTSD and chronic pain comorbidity following childhood sexual abuse. For example, Peles et al. [49] demonstrated CPTSD prevalence rates between 19.1% and 60% in female survivors of childhood sexual abuse receiving methadone maintenance treatment versus those without a history of opioid addiction. Chronic pain comorbidity rates differed between CPTSD versus non CPTSD patients (100% vs 50%) without a history of addiction. Tsur [50] investigated PTSD/CPTSD in association with trauma-related pain symptoms and found higher levels of CPTSD symptoms (i.e., PTSD + DSO) linked to higher rates of pain flashbacks (23.1%), which is considered a posttraumatic stress response centralizing around pain, compared to women reporting non-pain flashbacks (36.3%) and no flashback symptoms (40.6%). In both studies, chronic pain was a self-reported outcome based on the presence of persistent pain lasting for more than six months. None of the included studies in this review reported on CPTSD in clinically diagnosed chronic pain patients or those receiving care for pain management.

## Study objective 2: Interaction data between trauma factors and pain symptoms in individuals with chronic pain

**a. Childhood trauma, PTSD/CPTSD in individuals with chronic pain.** Except for two studies [49, 50], all included studies assessed childhood trauma in relation to PTSD as opposed to CPTSD. Several studies found that more severe childhood trauma, in particular maltreatment, was associated with PTSD in individuals with chronic pain [42, 48] when compared to those without PTSD and healthy controls [46]. For example, in a community sample, higher rates of childhood trauma exposure, including sexual abuse, were found in participants with IC/BPS and comorbid PTSD as opposed to those without PTSD, represented by medium to large effect sizes. No differences were found regarding adult trauma exposure, including physical and sexual abuse, between these groups [47]. As for evidence on CPTSD outcomes, Tsur [50] associated higher childhood sexual abuse severity with increased experiences of pain flashbacks as well as CPTSD symptoms compared to controls (i.e., non-pain flashbacks, no flashbacks).

**b. Trauma factors and pain symptoms in individuals with chronic pain.** Four studies included in this review explicitly investigated the association between childhood trauma, PTSD/CPTSD, and pain symptoms in individuals with chronic pain [41, 47, 49, 50]. For example, Coppens et al. [41] assessed childhood maltreatment in relation to perceived pain experiences and found an indirect effect of childhood abuse and neglect on both quantitative and qualitative pain reports through PTSD severity, representing medium effect sizes. No relationship between childhood maltreatment severity and pain reports was revealed, nor a moderator

effect of PTSD, suggesting a mediation effect. Other studies included in this review found direct effects of childhood trauma, in particular neglect and emotional abuse, on pain outcomes, including pain-related health impact and disability [39, 42].

McKernan et al. [47] investigated the role of criterion A trauma on the relationship between chronic pain phenotypes and PTSD. In a convenience sample of participants with IC/BPS and comorbid PTSD, higher rates of current pain and clinically relevant central sensitization (CS) were observed in individuals as opposed to those without PTSD, represented by medium to large effect sizes. When comparing IC/BPS subgroups based on CS levels, all patients with PTSD corresponded to criteria of the widespread IC/BPS phenotype, associated with higher rates of polysymptomatic complaints, psychosocial distress and pain levels. While IC/BPS participants with CS reported higher rates of childhood trauma as well as lifetime physical and sexual abuse, PTSD was shown to be uniquely related over and above trauma exposure to widespread pain phenotype of IC/BPS.

Another study, using quasi-experimental design assessed analgesic responses in FM patients with and without PTSD based on stress-induced changes in pain and intolerance thresholds during a Social Stress Test task [46]. Results revealed lower basal pressure pain and intolerance thresholds during recovery when compared to healthy controls, indicating hyper sensitivity at basal function in FM patients, regardless the presence of PTSD. In response to acute stress, however, FM patients showed differences in hypo reactivity during the task, such as a lack of hyperalgesic response in FM with PTSD during and after exposure as opposed to a delay of a hyperalgesic response in FM patients without PTSD. Higher childhood trauma severity was found in FM patients with PTSD than those without PTSD. Groups did not vary in pain intensity or chronicity levels of FM symptoms.

Regarding CPTSD, two studies investigated associations with chronic pain comorbidity in female survivors of childhood sexual abuse. For example, a cross-sectional study conducted in a clinical sample, demonstrated positive correlations between chronic pain symptoms (e.g., pain severity, number of painful body regions), sexual abuse-related PTSD and CPTSD severity in adulthood. Age of onset of first experience of sexual abuse was negatively associated with pain duration [49]. Another study provided evidence for understanding the link between childhood sexual abuse, CPTSD and chronic pain by highlighting the role of somatic pain-related manifestations of PTSD/CPTSD, in particular pain flashbacks. Further, results identified peritraumatic pain during childhood sexual abuse as a risk factor for chronic pain in adulthood [50]. Overall, both studies including CPTSD measurement highlighted high prevalence of chronic pain in survivors of childhood sexual abuse associated with higher psychiatric comorbidity, namely CPTSD.

Finally, two studies demonstrated transcultural validity for associations between childhood trauma, PTSD and chronic pain symptoms drawing from evidence obtained in clinical settings across Europe, North America, and the Middle-East [44, 45]. A study conducted in a community sample elucidated differences in chronic pain experiences in relation to abuse history based on sex differences [43]. Particularly, molestation was associated with higher affective pain, but only in men with chronic pain when compared with female participants. Similarly, childhood molestation predicted pain-related PTSD only in men, when controlling for race, sex and education. Female survivors of childhood sexual abuse were equally likely to have pain-related PTSD as women without a history of abuse.

## Study objective 3: Author's recommendations for future research and clinical practice

**a. Scientific research.**    In the study of etiology and pathophysiology of chronic pain, comorbid mental disorders and psychological distress should be considered [44]. Additional

research is also needed identifying mediating or moderating factors on the childhood trauma–HPA axis dysregulation relationship in chronic pain, using psychophysiological measures [48, 51]. Suggested characteristics of childhood trauma typically include developmental timing and subtypes, while calling for empirical attention to childhood neglect [45], as well as subsequent experiences of violence or abuse, and ongoing interpersonal relations later in life [48]. Concurrently, more attention should be addressed to pain-specific posttraumatic stress symptoms (e.g., pain flashbacks, avoidance of trauma-related pain sensations), as well as somatic manifestations of CPTSD in relation to chronic pain [50]. Future research should assess trauma focused-interventions in FM in order to further clarify trauma-based etiology of FM in comparison to other functional somatic syndromes, medically unexplained symptoms, somatic symptoms, and related psychopathology [42]. Some findings included in this review also warrant further investigation on whether some psychological states of detachment (e.g., dissociation) might explain hypo reactivity in FM patients as a coping strategy. When addressing trauma in the context of chronic pain, differences in patients based on the presence of PTSD should be considered in future research by using a differential profile approach [46]. Finally, in the study of abuse and trauma in relation to chronic pain, more research should include men [43].

**b. Clinical practice.** The majority of the included studies recommend systematic screening for trauma factors such as childhood trauma and PTSD/CPTSD [41, 42], regardless of race, age or gender [43]. Specific training might be needed to reduce identified barriers (e.g., lack of time, discomfort with subject, or lack of familiarity with the role of abuse) to appropriate and effective screening methods [43]. Screening procedures should also include detection for potential comorbid mental disorders in relation to abuse, such as somatoform dissociation disorder and alexithymia, using appropriate tools [44, 51]. Trauma-focused therapies may include Eye Movement Desensitization and Reprocessing (EMDR) [42], as well as intervention techniques based on Eccleston's model of tripartite system of threat protection in order to support FM patients with and without PTSD to engage in more adaptive stress responses [46]. As PTSD appears to be associated with the "widespread" pain phenotype, multimodal treatment should be considered for these patients [47]. Trauma-informed care is recommended in a more general way, emphasizing patient-care provider trust and rapport, reducing anxiety and increasing patient control and safety during appointments and medical examination procedures [47]. Finally, clinicians treating survivors of abuse should specifically inquire about chronic pain complaints, in order to facilitate tailored adequate approaches in comprehensive treatment [49].

## Discussion

Despite the growing evidence on the trauma-pain relationship, literature examining the association between childhood trauma and PTSD in relation to pain outcomes remains limited. This review further adds on existing systematic data by including evidence on CPTSD in individuals with chronic pain. In total, 13 studies were included in this systematic review. Study highlights have been summarized into the following sections in order to guide future research as well as recommended evidence-based clinical practice and policy in routine pain management.

### Childhood trauma: Neglect and emotional abuse in individuals with chronic pain

Different aspects of childhood trauma have been previously identified as risk factors for chronic pain conditions, such as nature of trauma [15, 52], and cumulative experiences of

maltreatment to [19, 53, 54]. In addition to existing systematic and metanalytical data, studies included in this review particularly emphasize the long-term consequences of emotional abuse and neglect as opposed to physical and sexual abuse. Consistent with DSM A-criterion type of traumatic events, other reviews typically focused on the impact of abuse specific childhood trauma (e.g., physical abuse, sexual abuse) [10, 52, 55, 56]. There is some research, however, indicating an independent relationship between PTSD symptoms and chronic pain outcomes following the presence of criterion A trauma history [57]. Moreover, present findings provide evidence for the expanded definition of trauma exposure by current PTSD/CPTSD ICD-11 guidelines, in particular with respect to the inclusion of childhood neglect and emotional abuse, in addition to DSM A criterion events. Despite suggested relevance to chronic pain etiology and PTSD/CPTSD comorbidity, research clarifying the differential impact of neglect and emotional abuse alongside events of childhood physical and sexual abuse remains minimal and warrants further investigation whether and to what extent these forms of trauma are associated with unique healthcare needs in chronic pain management.

## The long-term impact of childhood trauma: Evidence for differential patterns in PTSD/CPTSD and pain modulation processes

In total, only four studies included in this review explicitly investigated relationships between childhood trauma, PTSD/CPTSD and pain outcomes in individuals with chronic pain. The present findings are in accordance with other research demonstrating the negative impact of PTSD on pain outcomes when linked to childhood maltreatment compared to lower levels of pain typically experienced by individuals who have been diagnosed with PTSD alone [54, 58]. The long-term impact of cumulative childhood trauma was further recognized by an indirect dose-response relationship associated with increased risk of re-traumatization, higher levels of PTSD and perceived distress when compared to adulthood trauma. Similar to results of a recent systematic review [1], certain chronic pain phenotypes (e.g., "widespread pain") were identified as risk factors for described links.

This review also included evidence on biomarkers involved in pain modulation processes (e.g., cortisol secretion, pressure pain thresholds). In addition to existing systematic data [59], study findings support inhibitory capacity of adaptive allodynic responses in chronic pain patients with a history of childhood trauma by adding information to the role of PTSD. In this connection, differential neurophysiological patterns in chronic pain patients with PTSD compared to those without PTSD were associated with two main psychological/behavioral responses, namely hyperarousal and dissociation [46, 48]. This hypothesis is in line with previous studies, suggesting a unique paradoxical pain profile in individuals with chronic pain and PTSD, characterized by both pain-related hypo- and hyperresponsivity when compared to controls [8, 60]. Other research has emphasized the role of childhood versus adulthood trauma exposure in advancing current understanding of differential PTSD-related conditions (e.g., dissociation, depression) and physical health symptoms, including pain [61].

It is important to note, however, that results associating childhood trauma, PTSD, and pain are typically obtained in the absence of any CPTSD assessment. Only one study included in this review examined differential role of CPTSD symptoms in relation to childhood trauma, while identifying pain-related somatic manifestations (e.g., pain flashbacks) both as maintaining and worsening factors of chronic pain outcomes. These results are consistent with some preliminary research demonstrating associations between CPTSD symptoms (i.e., DSO symptoms) and higher rates of somatization [62] as well as abusive pain personification in individuals with childhood trauma compared to those with PTSD [33]. Despite important implications for empirical and clinical efforts as argued by a recent review [63], our understanding of

trauma-related bodily experiences remains an underdeveloped realm of translational pain research. In particular, findings in this review corroborate the current lack of validated and standardized assessment for pain-related trauma factors (e.g., peri and posttraumatic pain) which was identified as a major barrier to more robust methodological evidence. The need for future research adopting a differential analytical approach (e.g., cluster analysis), has also been issued to verify theorized relationships in order to extend current conceptual models of comorbidity and pain phenotypes by considering the unique features of CPTSD alongside PTSD symptoms.

## Trauma–pain comorbidity: Intersectional disparities

Although transcultural validity of trauma factors in chronic pain outcomes was consistently reported in this review [44, 45], the majority of the included study samples represented predominantly Caucasian and female individuals suffering from FM. Only one study provided some insight into intersectional disparities regarding childhood abuse in adults with chronic pain [43]. Findings corroborate the lack of available evidence identified by a recent review [64], emphasizing the critical need for more inclusive research to ensure that underrepresented groups receive equitable benefit from chronic pain research in terms of health and social policy. The same applies to trauma factors that remain oftentimes under-recognized, undertreated, or inadequately treated among marginalized groups [65]. More research is needed to explore the interplay of social factors (e.g., socioeconomic status, gender, race) and health disparities, while building on evidence for a more precise understanding of trauma-pain comorbidity and management within social context.

## Trauma focused treatment versus trauma-informed care

Considering the widespread prevalence of childhood trauma and both its long-term and complex impact on posttraumatic symptoms and pain related outcomes later in life, recommendations for clinical practice included in this review address the need for systematic screening of trauma factors in individuals seeking care for chronic pain. Consequently, psychotherapeutic strategies should target PTSD/CPTSD to relief illness burden, helping individuals with chronic pain to engage in more adaptive stress responses and promote general functioning [41, 42, 46]. Despite extensive literature on psychological treatment for PTSD, there is currently no "gold standard" for CPTSD screening or intervention methods. Furthermore, numerous limitations have been associated with first-line, evidence-based treatments for PTSD, including early dropout and worsening of symptoms in survivors of interpersonal trauma [66–68]. In this regard, Trauma Center Trauma Sensitive Yoga (TCTSY) [69]', an evidence-based protocol for complex trauma or treatment-resistant PTSD, appears to be a particularly promising therapeutic strategy, drawing specific focus to interoception (i.e., awareness of bodily sensations) and empowerment processes. While there is cumulative qualitative and quantitative evidence demonstrating protocol efficacy compared to conventional psychotherapy modalities [70–72], the use of TCTSY in individuals with chronic pain has not yet been investigated. In addition to trauma specialized treatment, and in line with a recent topical review [73], the present findings further support the importance of a systems approach to trauma care in pain management and rehabilitation services. Future research is needed to investigate comprehensive models of trauma-informed care based on principals such as safety, collaboration and choice within routine practice as a means to improve patient adherence, pain outcomes and prevent re-traumatization.

### Methodological considerations

This systematic review was conducted following recommended guidelines for search strategy as well as quality assessment allowing for a more rigorous process regarding methodological appraisal. Some limitations, however, should be taken into consideration in analyzing key findings. The search was not limited to study design, year of publication or methodological quality. Further, inclusion criteria for chronic pain and trauma factors were generally defined such as to provide a broad overview of the current state of art, limiting therefore conclusive or generalizing evidence regarding subtypes of trauma in relation to specific pain syndromes or phenotypes. Despite the inclusive approach to this review, only a short list of mostly moderate to high quality evidence, was identified, highlighting the preliminary nature of research in this area. Overall, the selected studies used appropriate and validated measurement for childhood trauma, PTSD/CPTSD and chronic pain which included a variety of self-reported as well as physician-based assessment. However, the heterogeneity of tools included in this review, in particular for PTSD/CPTSD, warrants vigilance to generalization of findings. Further, the majority of selected studies used the Childhood Trauma Questionnaire [CTQ; 74] as primary measurement for childhood trauma. While this is a validated and widely utilized instrument in the study of childhood trauma history, it provides assessment limited only to childhood maltreatment (i.e., abuse and neglect). Only two studies in this review assessed childhood trauma exposure based on PTSD qualifying stressors following DSM diagnostic criteria. This review recognizes the instability around diagnostic consensus of PTSD/CPTSD proposed by distinct classification models over the past two decades. For example, based on earlier diagnostic and clinical literature [22, 25], somatization was typically considered a core feature of DSM DES-NOS, but does not appear in the current WHO ICD-11 model of CPTSD. Finally, to the best of our knowledge, there is currently no randomized controlled, longitudinal or case study evidence investigating intervention modalities for PTSD/CPTSD and chronic pain comorbidity in individuals with a history of childhood trauma.

## Conclusion

The findings of this systematic review highlight the importance of taking into account childhood trauma, in particular neglect and emotional abuse, in the study of PTSD/CPTSD and chronic pain comorbidity in adults. The long-term impact of childhood trauma was further emphasized by an indirect dose-response relationship associated with increased risk of re-traumatization, higher levels of PTSD and perceived distress later in life when compared to adulthood trauma. This review also included evidence on specific neurophysiological patterns in chronic pain patients with PTSD suggesting differential pain modulation processes following trauma, in particular childhood maltreatment. Only a few selected studies reported on CPTSD and chronic pain comorbidity, providing preliminary evidence on the role of trauma-related physical pain (e.g., pain flashbacks). The need for future research adopting a differential approach has been issued in order to extend current models of comorbidity in relation to pain phenotypes, while also accounting for intersectional disparities. Considering the widespread prevalence of childhood trauma and its long-term and complex impact on both PTSD/CPTSD and pain chronicity later in life, recommendations for clinical practice draw attention to the need for PTSD/CPTSD specialized treatment as well as trauma-informed pain management in routine care.

## Supporting information

**S1 File. PRISMA checklist 2020.**
(DOCX)

**S2 File. List of identified studies in the literature search.**
(XLSX)

## Author Contributions

**Conceptualization:** Maria Karimov-Zwienenberg, Wilfried Symphor, Greg Décamps.

**Data curation:** Maria Karimov-Zwienenberg, Wilfried Symphor.

**Formal analysis:** Maria Karimov-Zwienenberg, Wilfried Symphor.

**Investigation:** Maria Karimov-Zwienenberg, Wilfried Symphor.

**Methodology:** Maria Karimov-Zwienenberg, Wilfried Symphor, William Peraud, Greg Décamps.

**Project administration:** Maria Karimov-Zwienenberg.

**Supervision:** Greg Décamps.

**Validation:** Maria Karimov-Zwienenberg, Wilfried Symphor, William Peraud, Greg Décamps.

**Writing – original draft:** Maria Karimov-Zwienenberg.

**Writing – review & editing:** Maria Karimov-Zwienenberg, Wilfried Symphor, William Peraud, Greg Décamps.

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
