## [Decision Letter · Decision Letter 0]

9 Jun 2024

PONE-D-24-10319Childhood trauma, PTSD/CPTSD and chronic pain: a systematic reviewPLOS ONE

Dear Dr. Karimov-Zwienenberg,

Thank you for submitting your manuscript to PLOS ONE. After careful consideration, we feel that it has merit but does not fully meet PLOS ONE’s publication criteria as it currently stands. Therefore, we invite you to submit a revised version of the manuscript that addresses the points raised during the review process.

Both reviewers have expressed interest in your manuscript, and I share their enthusiasm for your contribution to the literature. Both reviewers pointed to some aspects to improve your manuscript. Therefore, before accepting your work, I kindly request that you address each of the points raised by the reviewers and revise the concerning parts of your manuscript accordingly. In particular, there is a need to clarify the results of the search terms in order to guarantee replication and also to update the search.

We look forward to receiving your revised manuscript.

Kind regards,

Inga Schalinski

Academic Editor

PLOS ONE

“No external funding existed for the study. But we acknowledge financial publication support by Centre Hospitalier Agen-Nérac. The funders had no role in study design, data collection and analysis, decision to publish, or preparation of the manuscript.”

Additional Editor Comments:

Please be more cautious with use of “childhood trauma”, while referring to the definition of childhood adversities. This requires caution throughout the manuscript.

When using trauma I would recommend using the DSM-criteria of trauma exposure.

When referring to emotional abuse, physical abuse, sexual abuse and physical and emotional neglect I would recommend using “childhood maltreatment”.

Reviewers' comments:

Reviewer's Responses to Questions

**Comments to the Author**

1. Is the manuscript technically sound, and do the data support the conclusions?

Reviewer #1: Yes

Reviewer #2: Yes

2. Has the statistical analysis been performed appropriately and rigorously? 

Reviewer #1: N/A

Reviewer #2: Yes

3. Have the authors made all data underlying the findings in their manuscript fully available?

Reviewer #1: Yes

Reviewer #2: Yes

4. Is the manuscript presented in an intelligible fashion and written in standard English?

Reviewer #1: Yes

Reviewer #2: Yes

5. Review Comments to the Author

Reviewer #1: This review is interesting and timely. The link between trauma in childhood, PTSD and cPTSD and chronic pain conditions is important and summarising the available evidence to draw clinical perspectives is welcome.

Overall, the review is well structured. However, the manuscript could be improved. In some sections, especially toward the Discussion, the grammar and syntx of sentences were really not clear; with some mistakes that can be avoided (e.g. Quality of studies of the included studies could just read as Quality of the included studies.)

It was surprising to see the authors inspecting PROSPERO to make sure there was no other ongoing review n the topic, but why did not they register this systematic review in PROSPERO?

I was surprised by the relatively low number of records generated by the search. And when using their search strategy in Pubmed only, I received 13,646 results. Of course not all are relevant, but the discrepancy is massive. Could the authors explain how they ended up with only 297 records from heir search? Did the authors used tools like Covidence?

Maybe the authors will want to update their search while revising the manuscript?

What was the authors strategy to estimate the certainty of evidence if they did not use GRADE? The Joanna Briggs Institute (JBI) Critical Appraisal Checklist tool is dedicated to systematic reviews and research syntheses, not for empirical studies.

The description of the informations extracted from the publications is sometimes difficult to follow. I am sure the authors could do better to summarise the results, and could also include intermediate conclusions to help the reader to get the take home message after a long succession of summaries...

Study objective 3 confused me. Why not integrating this section to the Discussion where it would be more appropriate?

The discussion could end on a short conclusion section to summarise the overall take home message of the manuscript, as well as clinical and research perspectives.

Reviewer #2: I applaud the efforts of the author to bring light to the issue of a lack of rigorous scientific research in the area of childhood abuse as associated with trauma related conditions such as chronic pain and PTSD/CPTSD. I find the paper well written with the methodology easy to follow and comprehend. The results do not extend beyond the data and are appropriate. I only have a few suggestions to perhaps add to the limitations section:

1. Were the studies selected using appropriate assessment methods for BOTH chronic pain and PTSD? If so, it should be highlighted but, if not, it is definitely a limitation for future research to address.

2. Did any of the studies selected look at PTSD symptom cluster in association with childhood trauma. If so, it should be highlighted but, if not, it is another limitation for future research to address.

3. What about the PTSD symptoms of dissociation or depression? Research I've done in this area indicated the it is "... among women with childhood sexual or physical abuse, depression was the strongest predictor of somatic symptoms, whereas only dissociation predicted somatic symptoms among women without childhood sexual or physical abuse. Understanding the psychological and biological mechanisms that link PTSD comorbid depression or dissociation to physical health symptoms may aid development of individualized treatments for the physical and psychological consequences of trauma. In this context, a history of childhood trauma should also be taken into account." I recall seeing depression discussed in this review but dissociation should also be discussed or at least mentioned in the limitations section to be a part of assessment in future research.

Nevertheless, this paper, with the above additions, will make an important contribution to the literature and hopefully inform future research in this area.

6. PLOS authors have the option to publish the peer review history of their article (what does this mean?). If published, this will include your full peer review and any attached files.

Reviewer #1: No

Reviewer #2: No

---

## [Author Response · Author response to Decision Letter 0]

22 Jul 2024

Point-by-Point Rebuttal Letter

Journal requirements

Answer: We have checked the templates and made a few adjustments to meet the journal requirements, in particular with respect to headlines, Table titles, as well as file naming.

“No external funding existed for the study. But we acknowledge financial publication support by Centre Hospitalier Agen-Nérac. The funders had no role in study design, data collection and analysis, decision to publish, or preparation of the manuscript.”

Answer: None of the authors received material or financial funding for this study. However, the Centre Hospitalier Agen-Nérac agreed to pay the publication fee related to this article in PLOS ONE. 

Answer: Apart from the above-mentioned publication fee, there were no funders involved in the study project. 

Answer: None of the authors received salary from funders to conduct this research. 

The first author, Maria Karimov-Zwienenberg, is employee of the Centre Hospitalier Agen-Nérac and receives salary in the context of her clinical job which is unrelated to this research. 

Answer: “The authors received no specific funding for this work.”

Answer: The following statement has been added to the cover letter (See file: Cover Letter_Updated Statement):

“The authors received no specific funding for this work.” 

Answer: As requested, we hereby confirm that our submission contains all raw data required to replicate the results of this study. 

Additional Editor Comments

4. Please be more cautious with use of “childhood trauma”, while referring to the definition of childhood adversities. This requires caution throughout the manuscript. When using trauma I would recommend using the DSM-criteria of trauma exposure.

When referring to emotional abuse, physical abuse, sexual abuse and physical and emotional neglect I would recommend using “childhood maltreatment”.

Answer: Thank you for your comment. We revised the manuscript in order to avoid interchangeable use of terms regarding childhood adversity, maltreatment and trauma. For example (line 79, p.4), “Childhood trauma” was replaced by “Childhood adversity”. In addition, we decided to describe the nature of reported adverse or traumatic events by the included studies more explicitly within the Results section (objective 1, p 37-38). For example (line 227, p. 38): 

“Only two studies assessed childhood trauma exposure based on PTSD qualifying stressors following DSM criteria.”

In general, this review addresses, as mentioned in the Introduction section, the exposure of traumatic and adverse events in relation to PTSD and CPTSD through the lens of both DSM and ICD diagnostic criteria which provide distinct diagnostic conceptual frameworks. For example, following current ICD-11 guidelines, childhood trauma exposure has been recently expanded by including different types of interpersonal trauma, in particular childhood neglect and emotional abuse, in addition to DSM criterion A events. We chose to mention this more explicitly in the Introduction section (line 105, p.5). Further, consistent with recent data (e.g., Frewel et al., 2019; Cloitre et al., 2019) and earlier conceptual research in relation to complex PTSD (e.g., Van der Kolk; Cloitre et al., 2009), we decided to use childhood trauma for the purpose of this review as a general term when referring to the exposure to traumatic and adverse events before the age of 18. We chose to clarify this within the Introduction (line 121, p. 6):

“For the purpose of this review, in line with previous conceptual research and current ICD-11 PTSD/CPTSD guidelines, the term childhood trauma is used to address the exposure of traumatic or adverse events before the age of 18 years.”

Review Comments to the Author

5. Reviewer #1: This review is interesting and timely. The link between trauma in childhood, PTSD and cPTSD and chronic pain conditions is important and summarising the available evidence to draw clinical perspectives is welcome.

Answer: We thank the reviewer for this positive assessment.

5.1 Overall, the review is well structured. However, the manuscript could be improved. In some sections, especially toward the Discussion, the grammar and syntax of sentences were really not clear; with some mistakes that can be avoided (e.g. Quality of studies of the included studies could just read as Quality of the included studies.)

Answer: We thank the reviewer for this insight. We revised and modified the manuscript in order to improve grammar and syntax of sentences. Additionally, we corrected the following mistake “Quality of studies of the included studies” into “Quality of the included studies” (line 200, p. 35).

5.2. It was surprising to see the authors inspecting PROSPERO to make sure there was no other ongoing review on the topic, but why did not they register this systematic review in PROSPERO?

Answer: We understand this observation. As this research study was conducted as part of a dissertation project, it was initially uncertain whether a second reviewer would be available for data selection and extraction. When this was confirmed, the first author already started study selection and therefore the protocol did no longer meet eligibility criteria for registration with PROSPERO. We decided not to mention this detail within the manuscript. 

5.3. I was surprised by the relatively low number of records generated by the search. And when using their search strategy in Pubmed only, I received 13,646 results. Of course, not all are relevant, but the discrepancy is massive. Could the authors explain how they ended up with only 297 records from their search? Did the authors used tools like Covidence? Maybe the authors will want to update their search while revising the manuscript?

Answer: In fact, parenthesis “( )” were used for the search syntax. In order to promote data replicability, we added these within Table 1 of the manuscript (line 146, p. 7). Given the fact that this modification didn’t impact search results, we decided not to update the search. 

We did not use Covidence. Other software, namely Excel and Zotero, were used to streamline the process of conducting this systematic review. 

5.4. What was the authors strategy to estimate the certainty of evidence if they did not use GRADE? The Joanna Briggs Institute (JBI) Critical Appraisal Checklist tool is dedicated to systematic reviews and research syntheses, not for empirical studies.

Answer: We thank the reviewer for this comment. We did not use the JBI critical appraisal tool for systematic reviews. Instead, the methodological quality of each included study was assessed separately by using the corresponding design-specific critical appraisal checklist provided by JBI, including those for case-control studies, analytical cross-sectional studies, cohort studies and quasi-experimental studies (available from https://jbi.global/critical-appraisal-tools). We chose to address this information more explicitly in the Methods section (line 167, p. 11). 

In addition, based on previous systematic reviews using JBI critical appraisal checklist tools (e.g., Dijkshoorn et al., 2021; Whittaker et al., 2022), we decided to classify estimated levels of evidence as follows: 

- High quality studies: scores > 70%

- Moderate quality studies: scores between 50% and 70%

- Low quality studies: scores < 50%

This information was added in the Methods section (line 174, p.11). In order to enhance coherence within the manuscript, total quality scores were adjusted so to reference percentages instead of total scores in Table 5 (line 208, p 35).

5.5. The description of the informations extracted from the publications is sometimes difficult to follow. I am sure the authors could do better to summarise the results, and could also include intermediate conclusions to help the reader to get the take home message after a long succession of summaries...

Answer: We thank the reviewer for this suggestion. We revised the Results section (p. 36 – 44) in order to improve summary coherence and readability of extracted information. 

5.6. Study objective 3 confused me. Why not integrating this section to the Discussion where it would be more appropriate?

Answer: We thank the reviewer for this observation. Study objective 3 was articulated in response to the current lack of available translational and clinical research highlighted in the Introduction section. Resulting findings also add to existing data, as this, considering the literature, has not been systematically reviewed by previous research. We chose to reframe the third study objective (line 132, p. 6) to better distinguish results from study perspectives that are addressed in the Discussion section. 

5.7. The discussion could end on a short conclusion section to summarise the overall take home message of the manuscript, as well as clinical and research perspectives.

Answer: We thank the reviewer for this suggestion. A short conclusion was added to highlight main findings of the manuscript as well as recommendations for future research and clinical practice (line 546, p. 50)

6. Reviewer #2: I applaud the efforts of the author to bring light to the issue of a lack of rigorous scientific research in the area of childhood abuse as associated with trauma related conditions such as chronic pain and PTSD/CPTSD. I find the paper well written with the methodology easy to follow and comprehend. The results do not extend beyond the data and are appropriate. I only have a few suggestions to perhaps add to the limitations section. 

Answer: We thank the reviewer for this positive assessment.

6.1. Were the studies selected using appropriate assessment methods for BOTH chronic pain and PTSD? If so, it should be highlighted but, if not, it is definitely a limitation for future research to address.

Answer: Overall, the selected studies used appropriate and validated measurement for childhood trauma, PTSD/CPTSD and chronic pain which included a variety of self-reported as well as physician-based assessment. We added this information in the Methodological considerations section (line 528, p. 49) with some additional comments to consider in relation to PTSD and childhood trauma assessment. 

6.2. Did any of the studies selected look at PTSD symptom cluster in association with childhood trauma. If so, it should be highlighted but, if not, it is another limitation for future research to address.

Answer: Only one study selected in this manuscript reported PTSD cluster symptoms in a community sample of women with FM versus healthy controls (Ciccone et al., 2005). We chose to highlight this information in the Results section (line 252, p. 39).

6.3. What about the PTSD symptoms of dissociation or depression? Research I've done in this area indicated the it is "... among women with childhood sexual or physical abuse, depression was the strongest predictor of somatic symptoms, whereas only dissociation predicted somatic symptoms among women without childhood sexual or physical abuse. Understanding the psychological and biological mechanisms that link PTSD comorbid depression or dissociation to physical health symptoms may aid development of individualized treatments for the physical and psychological consequences of trauma. In this context, a history of childhood trauma should also be taken into account." I recall seeing depression discussed in this review but dissociation should also be discussed or at least mentioned in the limitations section to be a part of assessment in future research.

Answer: We thank the author for this insight. We decided to add this information in the Discussion section (line 459, p. 47) to highlight the importance of considering trauma childhood versus adulthood trauma exposure in relation to differential pathways in PTSD-related comorbidity and chronic pain.

---

## [Decision Letter · Decision Letter 1]

12 Aug 2024

Childhood trauma, PTSD/CPTSD and chronic pain: a systematic review

PONE-D-24-10319R1

Dear Dr. Karimov-Zwienenberg,

We’re pleased to inform you that your manuscript has been judged scientifically suitable for publication and will be formally accepted for publication once it meets all outstanding technical requirements.

Kind regards,

Inga Schalinski

Academic Editor

PLOS ONE

Additional Editor Comments (optional):

As you finalize the manuscript, please address the error on line 204 by removing the space before the comma

Reviewers' comments:

Reviewer's Responses to Questions

**Comments to the Author**

1. If the authors have adequately addressed your comments raised in a previous round of review and you feel that this manuscript is now acceptable for publication, you may indicate that here to bypass the “Comments to the Author” section, enter your conflict of interest statement in the “Confidential to Editor” section, and submit your "Accept" recommendation.

Reviewer #1: All comments have been addressed

2. Is the manuscript technically sound, and do the data support the conclusions?

Reviewer #1: Yes

3. Has the statistical analysis been performed appropriately and rigorously? 

Reviewer #1: N/A

4. Have the authors made all data underlying the findings in their manuscript fully available?

Reviewer #1: Yes

5. Is the manuscript presented in an intelligible fashion and written in standard English?

Reviewer #1: Yes

6. Review Comments to the Author

Reviewer #1: I have no additional comment, the authors have addressed all my comments. Well done, this was a great effort.

7. PLOS authors have the option to publish the peer review history of their article (what does this mean?). If published, this will include your full peer review and any attached files.

Reviewer #1: No

---

## [Editor Report · Acceptance letter]

21 Aug 2024

PONE-D-24-10319R1 

PLOS ONE

Dear Dr. Karimov-Zwienenberg, 

I'm pleased to inform you that your manuscript has been deemed suitable for publication in PLOS ONE. Congratulations! Your manuscript is now being handed over to our production team.

Kind regards, 

on behalf of

Dr. Inga Schalinski 

Academic Editor

PLOS ONE